# An elastic-plastic solution for the optimal thickness of a frozen soil wall considering an interaction with the surrounding rock

Lianfei Kuang[1☯], Pin-Qiang Mo[1,2☯]*, Kuan-Jun Wang[3], Bin Chen[4]

**1** State Key Laboratory for GeoMechanics and Deep Underground Engineering, School of Mechanics and Civil Engineering, China University of Mining and Technology, Xuzhou, Jiangsu, China, **2** R & D Center, Shenzhen Urban Public Safety and Technology Institute, Shenzhen, Guangdong, China, **3** Zhejiang Huadong Construction Engineering, POWERCHINA Huadong Engineering Corporation Limited, Hangzhou, Zhejinag, China, **4** MCC Underground Space Technology Research Institute, Wuhan Surveying Geotechnical Research Institute Co., LTD. of MCC, Wuhan, Hubei, China

☯ These authors contributed equally to this work.
\* pinqiang.mo@cumt.edu.cn

**Data Availability Statement:** All relevant data are within the manuscript.

**Funding:** KUANG: the Fundamental Research Funds for the Central Universities (Grant No.

## Abstract

The technology of artificial ground freezing has been widely applied in geotechnical engineering to support underground spaces, whereas the effects of excavation-induced large deformation and frictional and dilatant behavior of geomaterials are neglected in the current design. In this paper, a rigorous elastic-plastic solution of cavity contraction is proposed using a non-associated Mohr-Coulomb failure criterion to provide the optimal thickness of the frozen soil wall for excavation using artificial ground freezing technology, considering an interaction between the frozen soil wall and the surrounding soil/rock. After validation of a case study on a deep mine shaft against a numerical simulation, a thorough parametric study investigates the variation in the optimal thickness with the soil properties and initial stress conditions, as well as the effects of interaction and the critical condition. Compared to the existing solution, the proposed optimal thickness of the frozen soil wall is shown to contribute to both the design and cost-effectiveness in practical engineering, including tunneling and mine shaft construction.

## Introduction

Artificial ground freezing (AGF) technology has been widely applied in geotechnical engineering via the advanced refrigeration method [1–5]. The compressive strength and impermeability of a frozen soil mass are markedly increased as the in-situ pore water in the subsurface is frozen. The strengthened frozen soil acts as a cost-effective barrier, providing both earth support and groundwater cutoff during construction and soil excavation. Reliable and effective technology has been applied to many geotechnical problems, including vertical shafts, deep excavation, tunnels, groundwater control, structural underpinning, and containment of hazardous waste (e.g. Soo and Muvdi [6]; Quamruzzaman et al. [7]; Hong et al. [8]; Liu et al. [9]).

2020ZDPYMS18) MO: National Natural Science Foundation of China (Grant No. 51908546, Grant No. 52178374), China Postdoctoral Science Foundation (Grant No. 2020T130699) WANG: the National Natural Science Foundation of China (Grant No. 52108356) The funders had no role in study design, data collection and analysis, decision to publish, or preparation of the manuscript.

**Competing interests:** The authors have declared that no competing interests exist.

Cavity expansion/contraction theory is concerned with the changes in the stresses and displacements for soil elements around a cavity. Analytical solutions of cavity expansion and contraction have been proposed for geomaterials with many geotechnical implications [10–14]. These solutions facilitate analyses on the stability of underground structures, and serve as benchmarks for further developments and numerical simulations. The prediction of underground excavation-induced ground settlements has been developed by removing the initial stress of the cavity (i.e. cavity contraction), which contributes to the design of support systems to maintain stability and serviceability [11, 15–18].

Of particular interest in this paper is the response to the freeze wall-surrounding soil/rock system with excavation. When circular excavation is supported by a frozen soil wall (FSW), it is vital to understand the behavior of the wall-soil system during excavation and to provide an acceptable design with an optimal thickness of the frozen soil wall. The traditional elastic design formula is known as Lame's equation, which considers a frozen soil wall as an infinitely long hollow cylinder with a constant pressure at the outer boundary of the FSW. The behavior of frozen soil was assumed to be an elastic material with Tresca strength conditions, and the derived solution has been taken as the basis for many design theories of frozen soil walls. The elastic-plastic solution was first reported by Domke [19], allowing for a plastic region to develop in part of the frozen soil. A closed-form expression of the FSW thickness was also provided by a numerical technique when the Tresca criterion was adopted. As the frozen soil wall is embedded in the surrounding infinite soil or rock, the interaction between the FSW and the surrounding soil plays an important role in the performance of the ground freezing system. Yang et al. [20] proposed a mechanical model for excavation in FSWs that included an interaction with the surrounding rock. Although both frozen soil and surrounding rock were modeled as elastic, a closed-form solution of FSW thickness using Tresca or Mises strength conditions was provided, and the effect of the stiffness ratio was highlighted to confirm the influence of interaction. Additionally, the solution was extended in Yang et al. [21] for the consideration of elastic-plastic behavior based on Domke's critical condition. By assuming no volumetric strain in the plastic region of the FSW, the closed-form solution of the FSW thickness using the Mohr-Coulomb yield function was suggested, considering the excavation unloading effect and the interaction between FSW and the surrounding rock.

However, the current solutions neglect the effects of friction-induced dilation, which is typically significant to geomaterials. Large deformation caused by excavation is usually assumed to be infinitely small, which could result in large errors when estimating the optimal thickness of the FSW. The outer confining stress of the FSW is a changing parameter during construction, and plays a vital role in the FSW-soil interaction system. It is therefore necessary to improve the analytical method considering both the cohesive-frictional behavior of frozen soil and interactions between the FSW and the surrounding soil.

This paper attempts to provide a rigorous elastic-plastic solution of cavity contraction for the design of an optimal thickness of the frozen soil wall considering the interaction with the surrounding soil. For soil in the plastic region, a large strain analysis is incorporated within the solution to account for the excavation-induced large deformation, while the assumption of no volumetric strain is eliminated. The novelty of the developed solution lies in the extension of cylindrical cavity contraction for a cohesive-frictional material immersed in an elastic material mass, where the solution improves the precision of the existing methods for the design of the FSW. A numerical simulation is then conducted to validate the proposed solution, and a thorough parametric study is performed to investigate the influence of the soil properties and the initial stress conditions. The innovation of this study is also instantiated by the proposed optimal thickness of the frozen soil wall, aimed at providing efforts for the excavation design using ground freezing technology. The methodology adopted in this study is first elaborated with the

problem definition and the analytical solution of cavity contraction. The next section describes the results of a case study with validation, and a parametric study is then reported to examine the optimal thickness of the FSW.

## Methodology

### Problem definition

The problem of a frozen soil wall considering the interaction with the surrounding soil is modeled by an infinitely long hollow cylinder representing the frozen wall, which is surrounded by an unbounded geomaterial, as shown in Fig 1. Plane strain is assumed in terms of the infinitely long model, and the axis stress $\sigma_z$ is treated as the intermediate principal stress, as discussed in Yu and Houlsby [22]. The initial hydrostatic stress condition ($\sigma_{r,0} = \sigma_{\theta,0} = P_0$) is assumed to be homogeneous, neglecting the induced residual stresses during FSW formation. The radius of excavation in demand is $a_0$, whereas the frozen soil wall is designed with a radius of $b_0$. Therefore, the initial concentric-cylindrical soil layers include a frozen wall ($a_0 < r_0 < b_0$) and a surrounding soil ($b_0 < r_0 < +\infty$), where $r_0$ denotes the radius of a soil particle before cavity contraction.

The cavity pressure at $r_0 = a_0$ was reduced from $P_0$ to 0, representing the excavation process. At the end of the unloading process, the cavity radius is decreased from $a_0$ to $a$, and the wall-soil boundary is deformed from $b_0$ to $b$. Meanwhile, the plastic region is generated in the frozen soil wall, with the elastic-plastic boundary located at $r < c$ (see Fig 1). The behavior of the frozen wall is governed by the Mohr-Coulomb yield criterion, whereas the surrounding soil is assumed to be an elastic material obeying Hooke's law. Hence the material properties for the

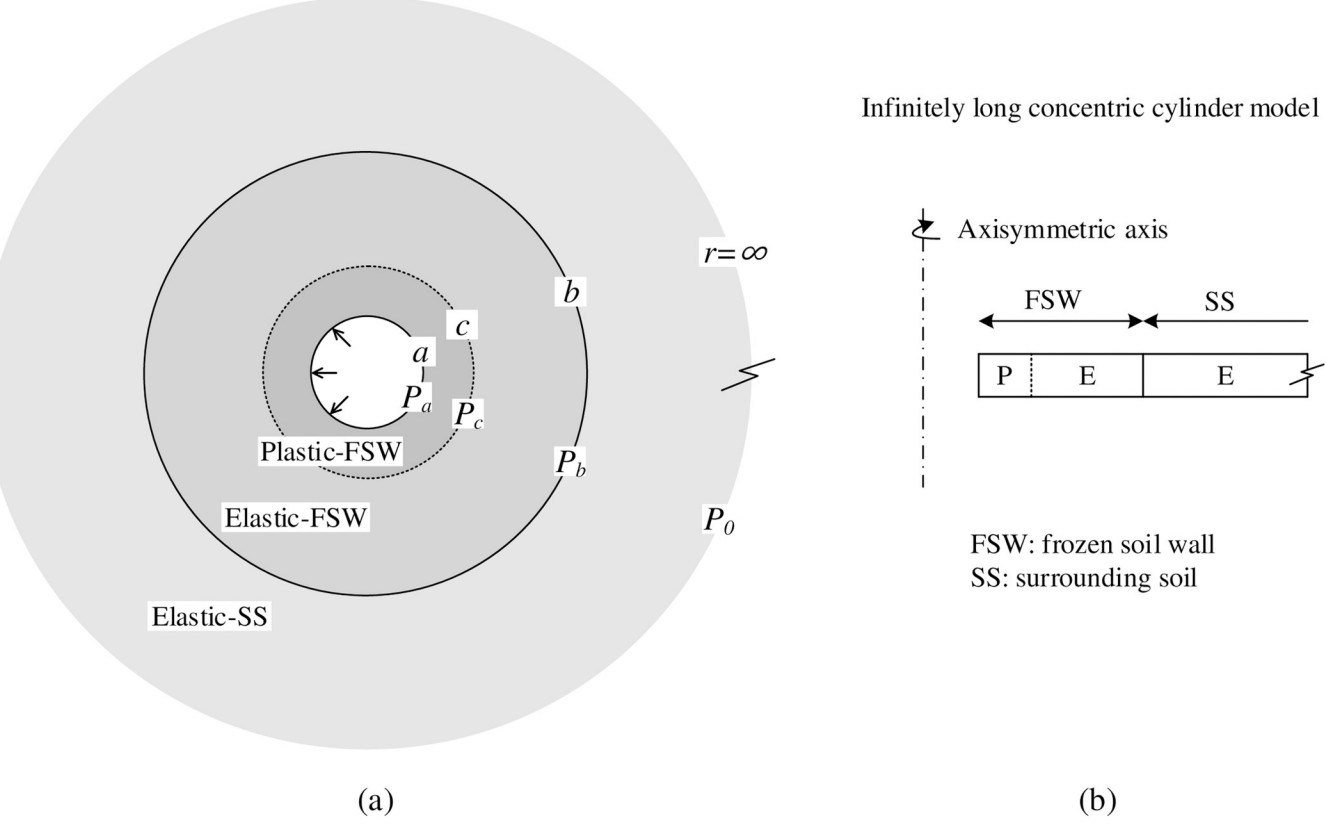

(a) (b)

**Fig 1. Infinitely long concentric cylinder model for cavity contraction.**

FSW include the Young's modulus $E_1$, Poisson's ratio $v_1$, cohesion $C_1$, friction angle $\phi_1$, and dilation angle $\psi_1$; the surrounding soil has Young's modulus $E_2$ and Poisson's ratio $v_2$ (the subscripts '1' and '2' represent the FSW material and the surrounding soil, respectively). Additionally, the following expressions are provided for mathematical convenience and for consistency with the solutions of Mo et al. [23]: $M_1 = \frac{E_1}{1-v_1^2}$; $M_2 = \frac{E_2}{1-v_2^2}$; $Y_1 = \frac{2C_1\cos\phi_1}{1-\sin\phi_1}$; $\alpha_1 = \frac{1+\sin\phi_1}{1-\sin\phi_1}$.

Note that tension positive is used throughout this paper. The small strains in the elastic regions can be expressed in terms of the radial displacement ($u$) as follows:

$$\varepsilon_r = \frac{du}{dr}; \ \varepsilon_\theta = \frac{u}{r} \tag{1}$$

For a soil particle in the plastic region of FSW, logarithmic strain components are adopted to account for the large strain around the cavity wall [22, 23], namely:

$$\varepsilon_r = \ln\left(\frac{dr}{dr_0}\right); \ \varepsilon_\theta = \ln\left(\frac{r}{r_0}\right) \tag{2}$$

## Analytical solution of cavity contraction

Cavity expansion in cohesive-frictional soils with a large strain analysis was developed in Yu and Houlsby [22], and the solution was extended by Mo et al. [23] for cavities embedded in two concentric layers of Mohr-Coulomb materials. Yu and Houlsby [24] proposed a large strain solution for contraction from an elastic-plastic stress state for cavities in an unbounded Mohr-Coulomb soil, while the solution for contraction from an in-situ stress state was provided by Yu and Rowe [17]. In this paper, the developed solution of cavity contraction, modified based on Mo et al. [23] and Yu's work, is presented for a cylindrical cavity within two concentric layers of frozen soil walls and surrounding soil as illustrated in Fig 1. With radial symmetry conditions, the stresses of the soil around a cavity satisfy the following equilibrium equation:

$$\sigma_\theta - \sigma_r = r\frac{\partial\sigma_r}{\partial r} \tag{3}$$

## Elastic regions in the surrounding soil and the frozen soil wall

**Surrounding soil ($r \geq b$).** The solution of the cavity contraction in the elastic regions has been derived following Mo et al. [23] in terms of the unloading effect. Note that the elastic solution of cavity contraction is identical to that of cavity expansion. The displacement distribution in the surrounding soil is expressed in the form as follows:

$$u = \frac{b(b - b_0)}{r} \tag{4}$$

Radial and tangential stresses, subject to the boundary conditions, can thus be shown as:

$$\sigma_r = \frac{M_2}{1 - \left(\frac{v_2}{1-v_2}\right)^2}\left[\left(\frac{v_2}{1-v_2} - 1\right)\frac{b(b-b_0)}{r^2}\right] - P_0$$

$$\sigma_\theta = \frac{M_2}{1 - \left(\frac{v_2}{1-v_2}\right)^2}\left[\left(1 - \frac{v_2}{1-v_2}\right)\frac{b(b-b_0)}{r^2}\right] - P_0 \tag{5}$$

**Frozen soil wall ($c \leq r \leq b$).** Accordingly, the displacement distribution in the elastic region of FSW is of the form:

$$u = D_1 r + \frac{D_2}{r} \tag{6}$$

where $D_1 = \frac{(c-c_0)c - (b-b_0)b}{c^2 - b^2}$ and $D_2 = \frac{(c_0 b - c b_0)cb}{c^2 - b^2}$ (after Mo et al. [23]).

The radial and tangential stresses are then expressed as:

$$\sigma_r = \frac{M_1}{1 - \left(\frac{v_1}{1-v_1}\right)^2} \left[ \left(D_1 - \frac{D_2}{r^2}\right) + \frac{v_1}{1-v_1}\left(D_1 + \frac{D_2}{r^2}\right) \right] - P_0$$

$$\sigma_\theta = \frac{M_1}{1 - \left(\frac{v_1}{1-v_1}\right)^2} \left[ \frac{v_1}{1-v_1}\left(D_1 - \frac{D_2}{r^2}\right) + \left(D_1 + \frac{D_2}{r^2}\right) \right] - P_0 \tag{7}$$

**Plastic region in the frozen soil wall ($a \leq r \leq c$).** As the soil is modeled as an isotropic dilatant elastic-perfectly plastic material, the Mohr-Coulomb yield criterion for the unloading scenario leads to the following expression:

$$\alpha_1 \sigma_r - \sigma_\theta = Y_1 \tag{8}$$

Combining the equilibrium equation (Eq 3) and the yield function (Eq 8) gives the expressions of the radial and tangential stresses:

$$\sigma_r = \frac{Y_1}{\alpha_1 - 1} + A_1 r^{\alpha_1 - 1}$$

$$\sigma_\theta = \frac{Y_1}{\alpha_1 - 1} + A_1 \alpha_1 r^{\alpha_1 - 1} \tag{9}$$

where $A_1 = -\left(P_c + \frac{Y_1}{\alpha_1 - 1}\right)c^{1-\alpha_1} = -\frac{Y_1}{\alpha_1 - 1}a^{1-\alpha_1}$, based on the boundary conditions $(\sigma_r|_{r=c} = -P_c; \; \sigma_r|_{r=a} = -P_a = 0)$.

The following relationship for the plastic radius in terms of the cavity radius can therefore be obtained:

$$\left(\frac{c}{a}\right)^{1-\alpha_1} = \frac{\frac{Y_1}{\alpha_1 - 1}}{P_c + \frac{Y_1}{\alpha_1 - 1}} \tag{10}$$

Due to the theory of plasticity, strains in the plastic region can be decomposed into elastic and plastic components, where the elastic strains obey Hooke's law. Plastic strains can therefore be determined through the non-associated Mohr-Coulomb flow rule for the contraction of cavities [17]:

$$\frac{\dot{\varepsilon}_r^p}{\dot{\varepsilon}_\theta^p} = \frac{\dot{\varepsilon}_r - \dot{\varepsilon}_r^p}{\dot{\varepsilon}_\theta^p} = -\beta_1 \tag{11}$$

where $\beta_1$ is a function of dilation angle ($\beta_1 = \frac{1+\sin\psi_1}{1-\sin\psi_1}$), and the associated flow rule can be achieved by selecting $\psi_1 = \phi_1$ (i.e. $\beta_1 = \alpha_1$).

By using the large strain assumption, the techniques of transformation and integration (after the approach of Mo et al. [23]) lead to the following expression, with the aid of a series

expansion:

$$\frac{\chi_1}{\gamma_1}(-A_1)^{\gamma_1}\left(c_0^{\beta_1+1}-a_0^{\beta_1+1}\right)=\sum_{n=0}^{\infty}\frac{\mu_1^n}{n!(n+\gamma_1)}\left[\left(P_c+\frac{Y_1}{\alpha_1-1}\right)^{n+\gamma_1}-\left(\frac{Y_1}{\alpha_1-1}\right)^{n+\gamma_1}\right] \quad (12)$$

where

$$\chi_1=\exp\left[\frac{1}{M_1}\frac{1-2v_1}{1-v_1}(\beta_1+1)\left(P_0+\frac{Y_1}{\alpha_1-1}\right)\right]$$

$$\mu_1=\frac{1}{M_1}\left[1+\beta_1\alpha_1-\frac{v_1}{1-v_1}(\beta_1+\alpha_1)\right] \quad (13)$$

$$\gamma_1=\frac{\beta_1+1}{\alpha_1-1}$$

**Solution procedure for the optimal thickness of FSW.** The critical condition for the frozen soil wall is assumed based on Domke's approach, which indicates that the size of the elastic-plastic boundary has the following relation:

$$c_0=\sqrt{a_0\cdot b_0} \quad (14)$$

To provide the optimal thickness of FSW for excavation with a radius of $a_0$, the following procedure might be adopted based on the necessary aforementioned equations: (i) for a given value of $c_0$, the size of FSW $b_0$ can be determined from Eq (14); (ii) the boundary conditions of the stresses at $r=b$ and $r=c$ (Eqs 5, 7 and 8) lead to the solutions of $P_b$, $P_c$, $b$, and $c$; (iii) the cavity radius after contraction $a$ is thus calculated from Eq (10), and a return value of $c_0$ is then determined based on Eq (12); (iv) an iteration process of the first three steps provides the value of $c_0$ and thus the optimal thickness of FSW (i.e. $b_0-c_0$).

It is noted that the proposed solution is applicable to all soil types, and that the mechanical responses of the unfrozen and frozen materials are characterized by the Mohr-Coulomb failure criterion. To assist the formulation based on the aforementioned derivations, a flowchart of the proposed analytical solution is provided in Fig 2. A MATLAB-based code is then programmed to calculate the distributions of stresses and strains after cavity contraction, and to provide the optimal thickness of the frozen soil wall for excavation.

## Results and discussion

### Results of a case study

The method of freeze sinking has been widely used in China to improve the soil strength and shaft wall stability during excavation (e.g. Wang et al. [25]; Liu and Wang [26]; Zhao et al. [27]). Systematic studies on the mechanical properties of frozen soils have been reported, which investigate the influences of temperature, stress condition, and the moisture content [28–32]. A case study from a coal mine site in Shandong, China is selected in this research to analyze the stability of the FSW using the proposed solution. Fig 3 shows typical results of a triaxial test for a sample of frozen clayey soil from this site at -20˚C under a high confining stress of 10 MPa, after Leng [33]. It is noted that both the shear strength and the back-calculated Yong's modulus of the frozen soil are higher than typical values of clayey soils, which is attributed to the high confining pressure and the low temperature for simulating frozen soil of an FSW at a great depth.

The results for this case study are provided first to describe the distributions of stresses and strains after cavity contraction, with numerical simulation comparisons. The engineering case is chosen after Yang et al. [21], which is a sinking shaft in the Tertiary thick clay layer using the

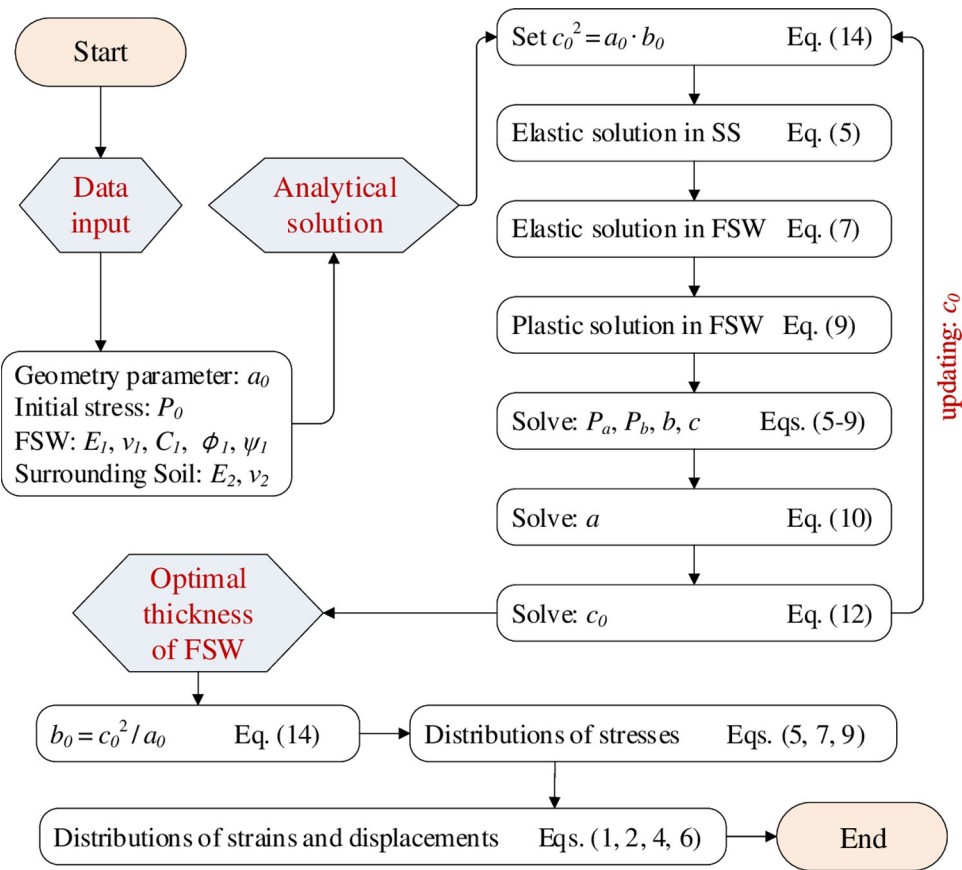

**Fig 2. Flowchart of the analytical solution for calculation of the optimal thickness of the FSW.**

ground freezing technique, where the average temperature of the frozen soil wall was approximately −20˚C. According to the Unified Soil Classification System [34], the soil is clayey soil with a low plasticity (CL). As the soil surrounding a deep vertical shaft is typically embedded at a large depth, the compacted hard soil usually behaves with higher modulus and ultimate strength values. Moreover, the freezing process of the AGF can significantly improve the stiffness and strength of the soil.

According to the data of experimental tests by Yang et al. [18], the soil parameters of FSW were reasonably set as: $E_1$ = 300 MPa, $v_1$ = 0.2, $\phi_1 = \psi_1 = 8˚$, $C_1$ = 3.5 MPa; the elastic parameters of the surrounding soil were then selected as $E_2$ = 100 MPa and $v_2$ = 0.2. The calculation is focused on the shaft at a depth of 500 m, which has a mean stress condition of $P_0$ = 6.5 MPa (after Yang et al. [21]).

The proposed analytical solution provides that the optimal thickness of the FSW is $b_0/a_0$ = 2.0045 and that the resultant cavity wall is $a/a_0$ = 0.9547. After the removal of the cavity pressure from $P_0$, the distributions of the stresses, strains and displacement of the soil elements around the cavity are shown in Fig 4. The radial stress is decreased to zero at the cavity wall, while the tangential stress is increased by cavity contraction. Although the mean stress in the surrounding soil is equivalent to $P_0$, a discontinuous tangential stress is found at the soil-wall boundary, and the tangential stress transfers to a reduction in the plastic region of FSW. Similarly, the radial strain increases and the tangential strain decreases during the unloading process. Note that the volumetric strain ($\varepsilon_r + \varepsilon_\theta$) is not zero in the area of the frozen soil wall.

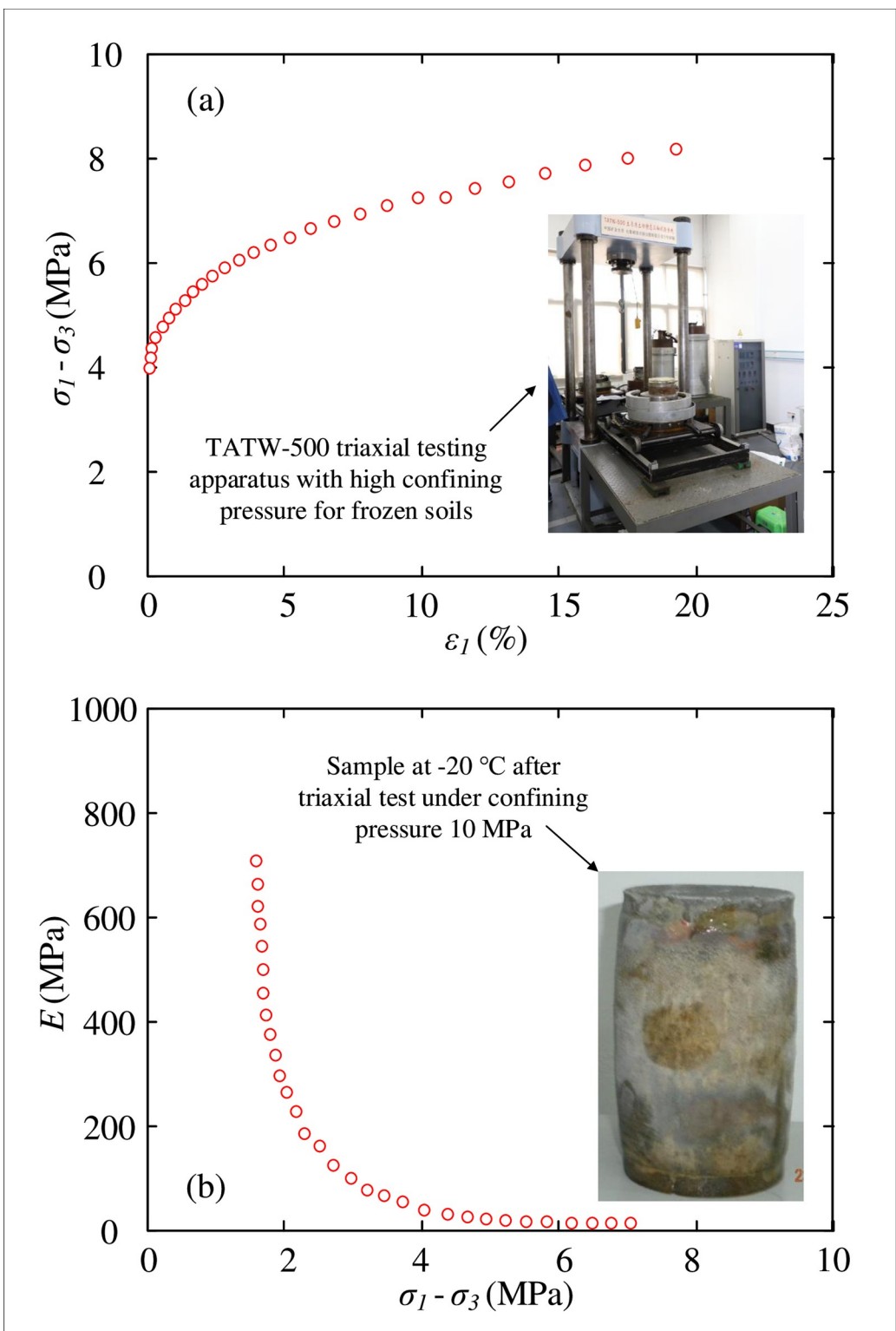

**Fig 3. Typical results of a triaxial test for a sample of frozen soil at -20˚C and 10 MPa stress condition, after Leng [33].**

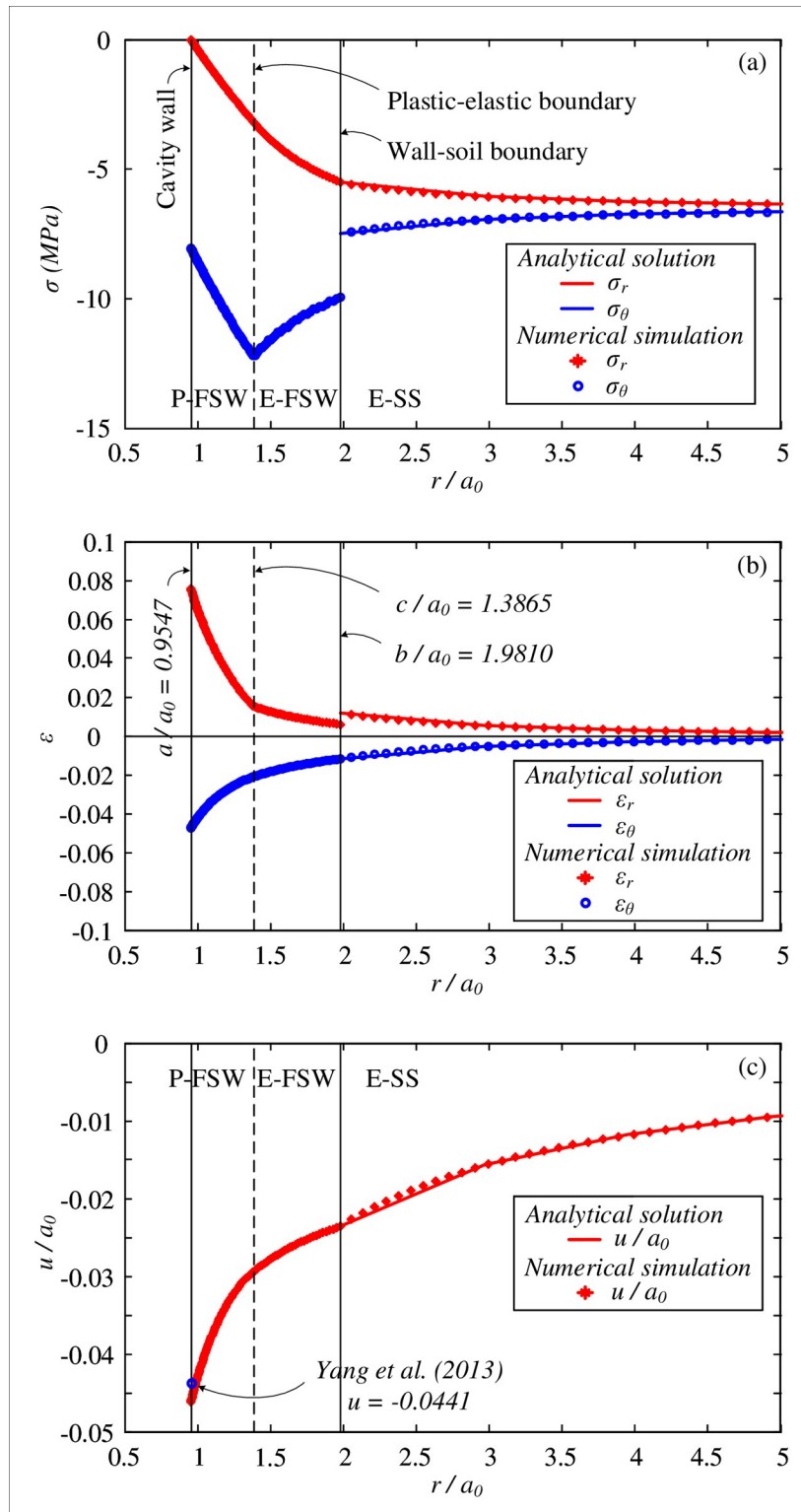

**Fig 4.** Distributions of (a) stresses; (b) strains; and (c) displacement, with comparisons of numerical simulation.

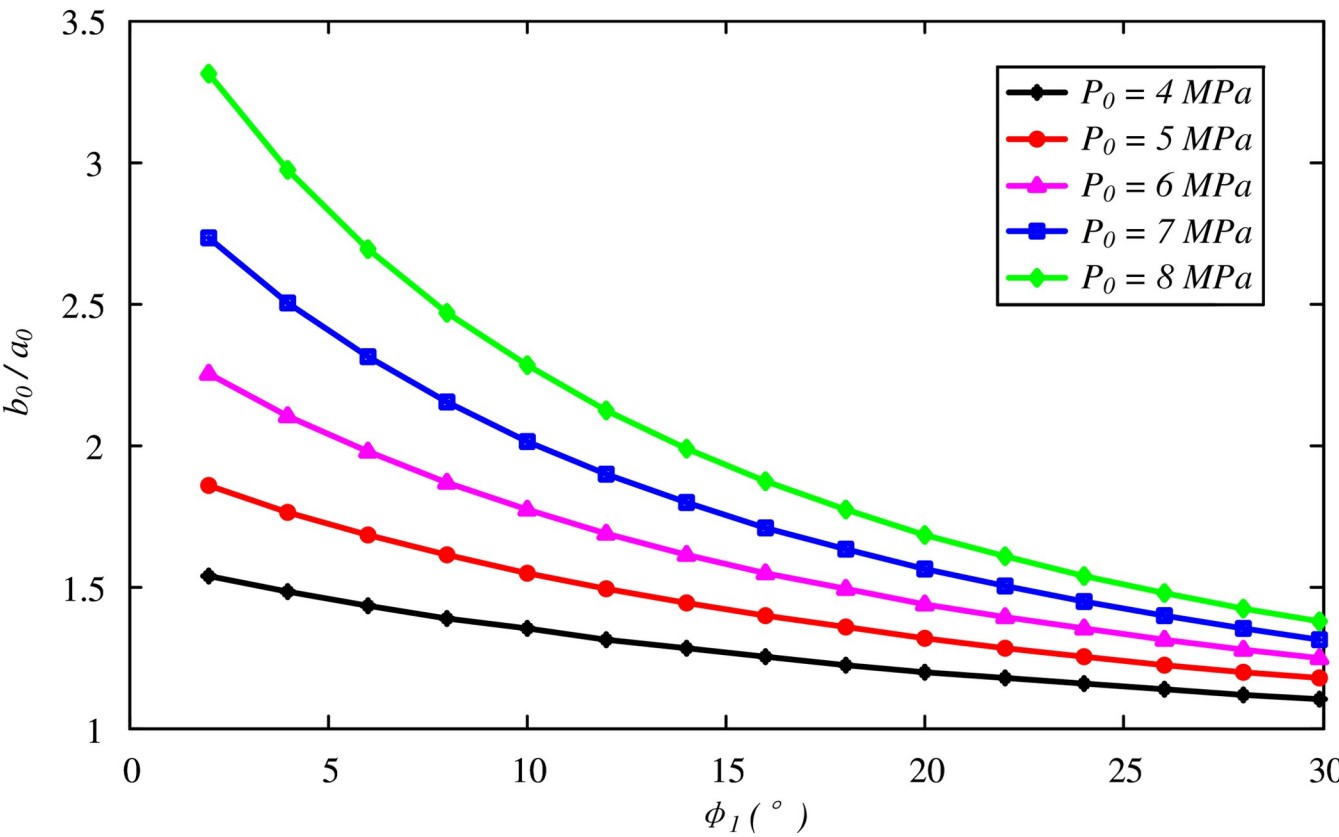

**Fig 5. Variation of optimal thickness of the FSW with friction angle $\phi_1$.**

Numerical simulation using the Abaqus-based finite element model with the calculated optimal thickness of the FSW ($b_0/a_0 = 2.0045$) was conducted to validate the proposed analytical solution, as also shown in Fig 4. The developed elastic-plastic solution of cavity contraction is confirmed by the good agreement with the numerical results. Despite the mesh size in the numerical simulation, the plastic boundary after excavation is $c_0/a_0 = 1.4114$, compared with $c_0/a_0 = 1.4158$ from the analytical solution. However, the optimal thickness was suggested to be $b_0/a_0 = 2.0959$ from the solution by Yang et al. [21] (small strain and zero volumetric strain assumptions in the plastic region), and the calculated cavity contraction $a/a_0 = 0.9559$ leads to a cavity wall displacement of $u = -0.0441$, which is smaller than that of this study and the numerical simulation (see Fig 4C). It is therefore believed that the proposed solution considering a large strain provides a more precise analysis of the soil behavior around the cavities.

## Parametric study

A thorough parametric study is provided in this section to investigate the effects of soil properties on the optimal thickness of the FSW. The aforementioned case study was taken as a reference to determine the soil properties. The initial hydrostatic stress condition varies between 4 and 8 MPa. Fig 5 shows the variation in the optimal thickness with the friction angle $\phi_1$. Note that the effect of the dilation angle is not included by using $\psi_1 = \phi_1$, indicating the associated flow rule of the Mohr-Coulomb model. It is determined that $b_0/a_0$ decreases nonlinearly with

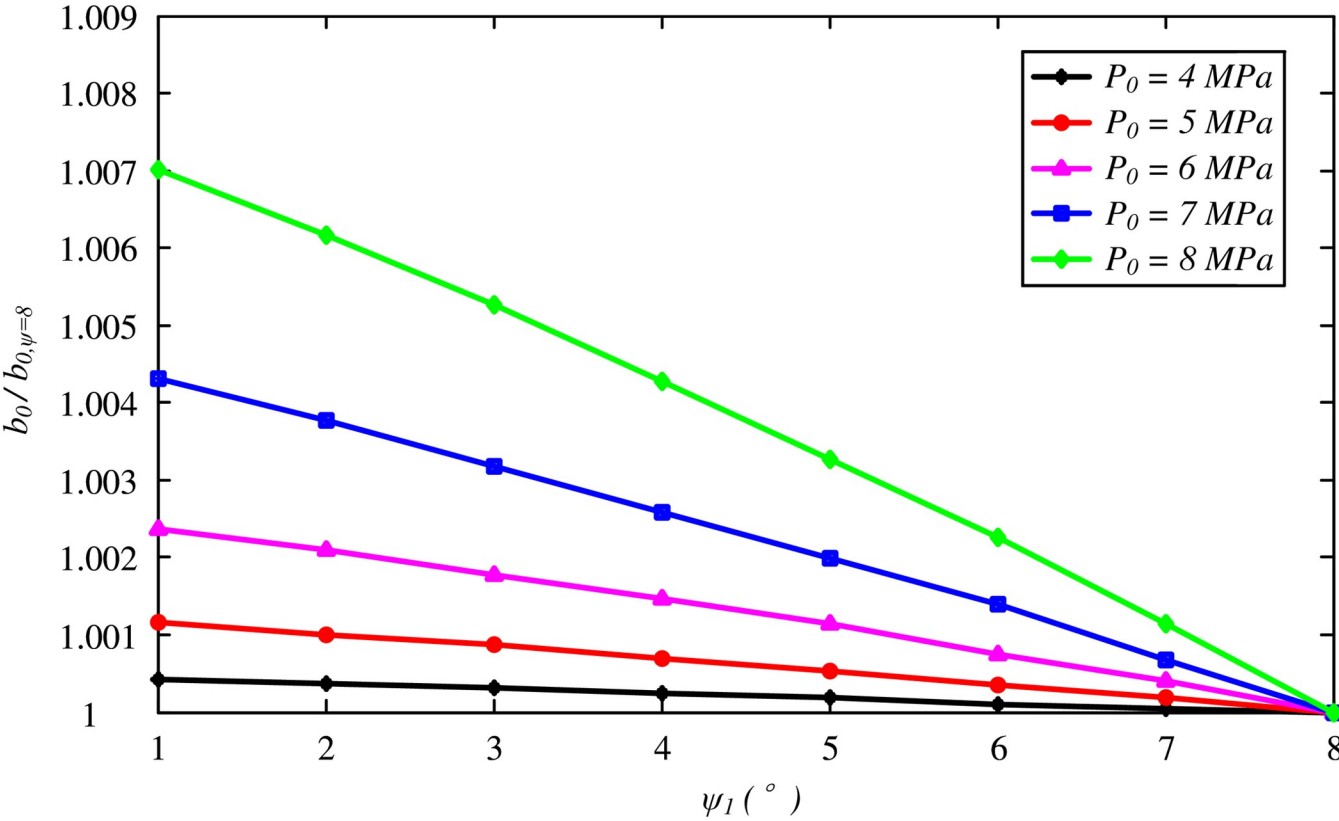

**Fig 6. Variation of optimal thickness of the FSW with dilation angle $\psi_1$.**

$\phi_1$, and a thicker layer of the FSW is required for higher initial stress conditions. The effect of the dilation angle is also presented in Fig 6 for a constant friction angle value $\phi_1 = 8˚$. Although a smaller dilation angle results in a larger FSW thickness, the effect is likely to be negligible.

Fig 7 investigates the effect of the cohesion of the FSW on the optimal thickness, with $C_1$ varying between 0.35 and 35 MPa. The value of $b_0/a_0$ is significantly increased with a decreasing cohesion of frozen soil. With a smaller value of $C_1$, the effect of $P_0$ is reduced and even reversed for $C_1 = 0.35$ MPa, where the higher stress condition results in a smaller value of the optimal thickness. Additionally, the effect of the Young's modulus is shown in Fig 8 by varying $E_1/E_2$ from 1 to 10. The optimal thickness increases with the FSW stiffness, whereas the increasing effect becomes small for $E_1/E_2 > 10$, as was reported by Yang et al. [21].

## Effects of the interaction and critical condition

The analytical solution provides an analysis of the unloading process considering the interaction between the frozen soil wall and the surrounding soil, which acts as a more reasonable model compared to the traditional Domke's model, and may reduce the designed thickness of the frozen soil wall. The solution for the model without an interaction can be achieved by either simplifying the proposed solution or choosing the parameter $E_2$ to be infinitely small. In this way, the interaction effect is presented in Fig 9 ($P_0 = 6.5$ MPa was used for this test), where

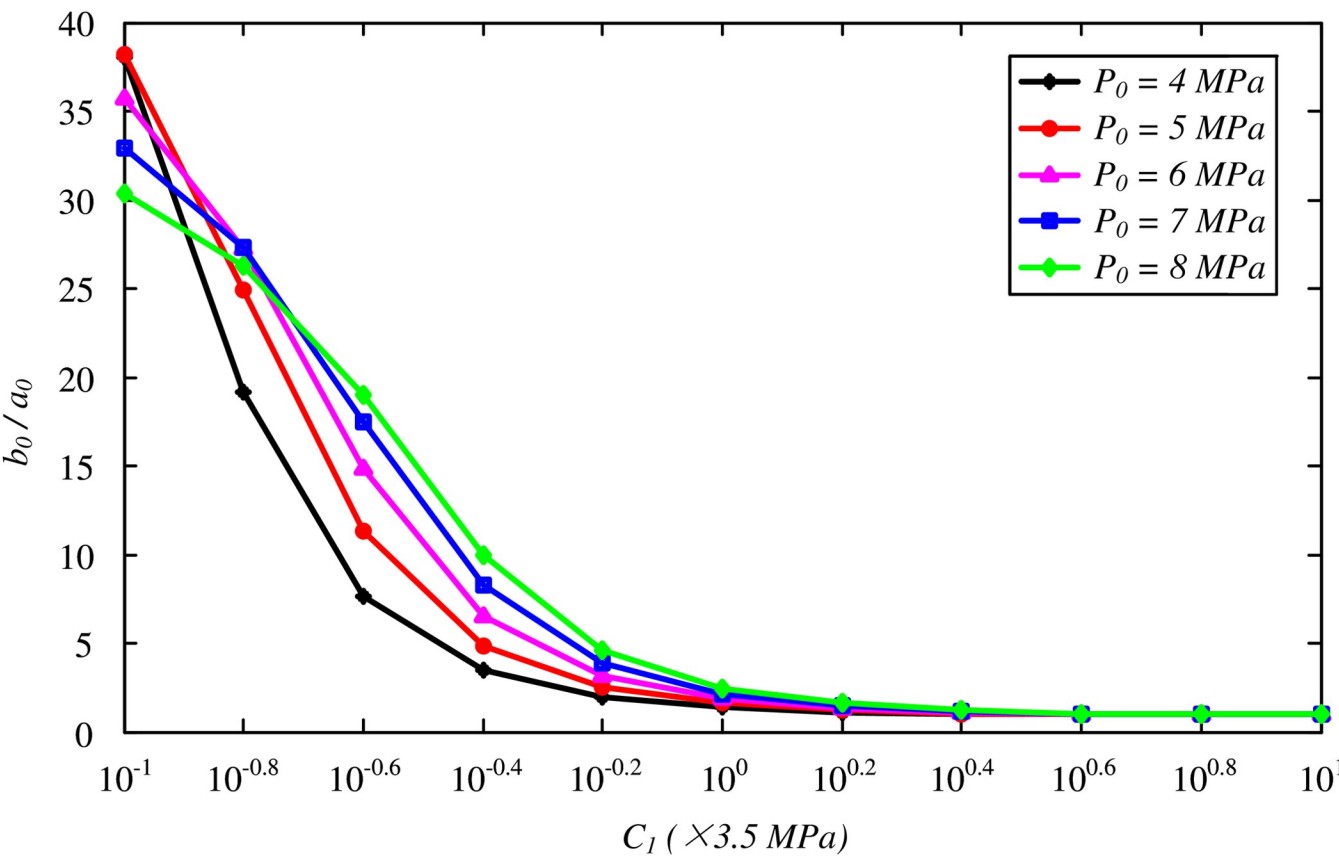

**Fig 7. Variation of optimal thickness of the FSW with cohesion $C_1$.**

the optimal thickness for the model considering the interaction is approximately 12% ~ 15% smaller than that of the model without the interaction. Additionally, Yang et al. [21] reported a closed-form solution of the optimal thickness of an FSW considering an interaction, as well as an additional assumption of no volumetric strain in the plastic region. The current exact solution is found to provide a smaller optimal thickness by approximately 2.5% ~ 5.5%, which may contribute to both the design and cost-effectiveness in practical engineering.

As mentioned in the solution, the elastic-plastic solution for the optimal thickness of the FSW is based on the critical condition after Domke's approach (Eq 14). The effect of the critical condition is investigated in Fig 10, as a reference for the further design of the FSW. Testing parameters were chosen according to Yang et al. [20], namely: $E_1 = 300$ MPa, $v_1 = 0.3$, $C_1 = 2.65$ MPa, $E_2 = 100$ MPa, $v_2 = 0.2$, $P_0 = 2.6$ MPa (about 200 m depth). When replacing the critical condition by $c_0 = a_0$, the solution is recovered to an elastic solution with a yield condition at the cavity wall, and the optimal thickness of the frozen soil wall is consequently increased to sustain the unloading process, as shown in Fig 10A. The closed-form solution for the critical condition of $c_0 = a_0$ can therefore be provided as:

$$b_0 = b\left[1 - \frac{P_0 - P_b}{M_2(v_2 - 1)}\right] \qquad (15)$$

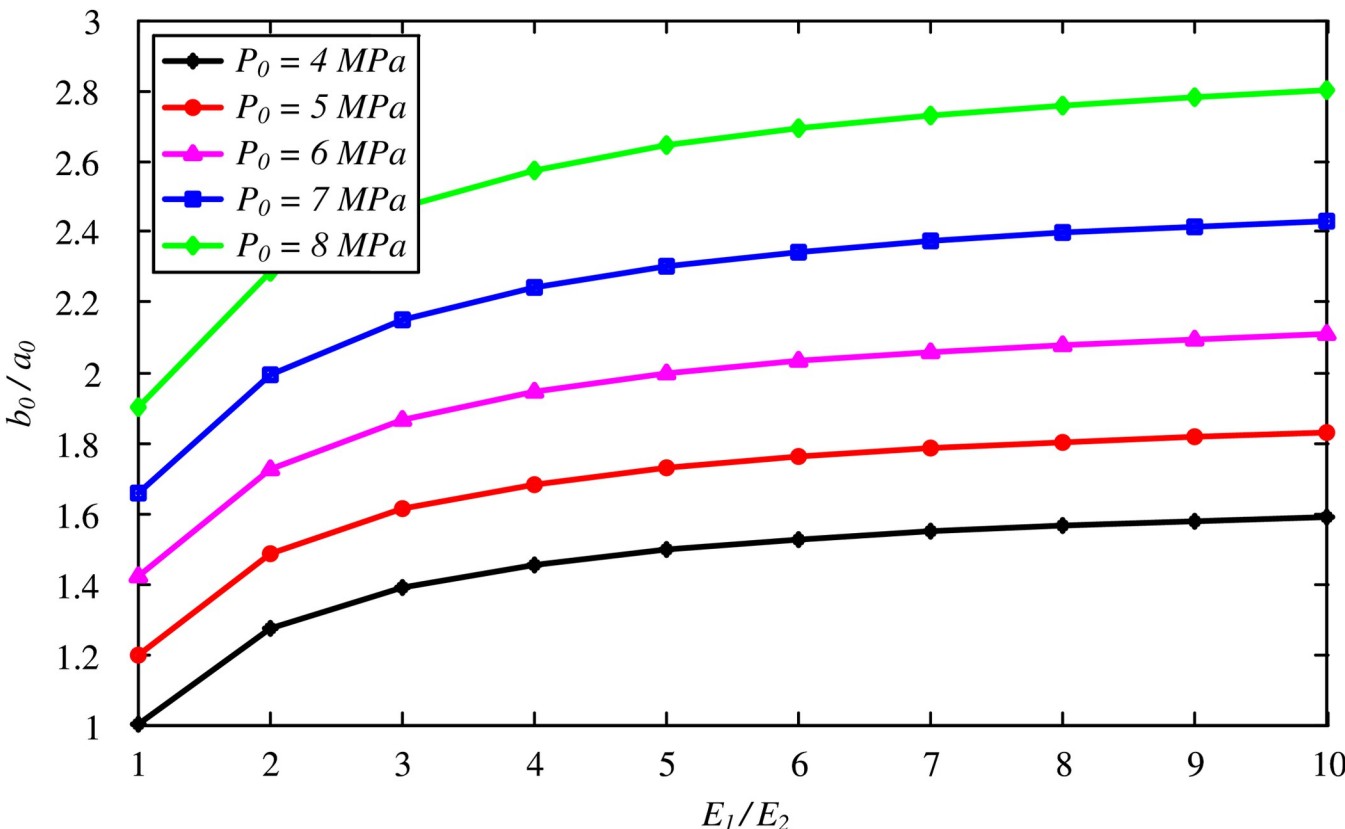

**Fig 8. Variation of optimal thickness of the FSW with stiffness $E_1$.**

where

$$P_b = \frac{\dfrac{(2P_0 - Y_1)(1 - 2v_1)}{2M_1(1 - v_1)} + \dfrac{Y_1}{2M_1(v_1 - 1)} - \dfrac{P_0}{M_2(v_2 - 1)}}{\dfrac{1}{M_1(v_1 - 1)} - \dfrac{1}{M_2(v_2 - 1)}}$$

$$b = a\sqrt{\frac{1}{1 - 2P_b/Y_1}} \tag{16}$$

$$a = \frac{a_0}{1 - \dfrac{P_b}{M_1(v_1 - 1)} - \dfrac{P_0 - P_b}{M_2(v_2 - 1)}}$$

The optimal thickness of the FSW is significantly increased with the reduction of the friction angle. Note that the Tresca model can be recovered when the friction angle is set to zero. The solution of Yang et al. [20] was presented for the elastic analysis of the frozen soil wall with an interaction with the surrounding soil using the Tresca and Mises criteria. By comparison, the Tresca strength is assumed to be $\sigma_c = 2C_1 = 5.3$ MPa, and the calculated optimal thickness is recovered for the test with $\phi_1 = 0°$. When the critical condition is extended to allow for a fully plastic region of the frozen soil wall (i.e. initial state when $c_0 = b_0$), the optimal thickness is then reduced by 10% for $\phi_1 = 2°$ and 2% for $\phi_1 = 30°$, as shown in Fig 10B. It is shown that a proper critical condition is

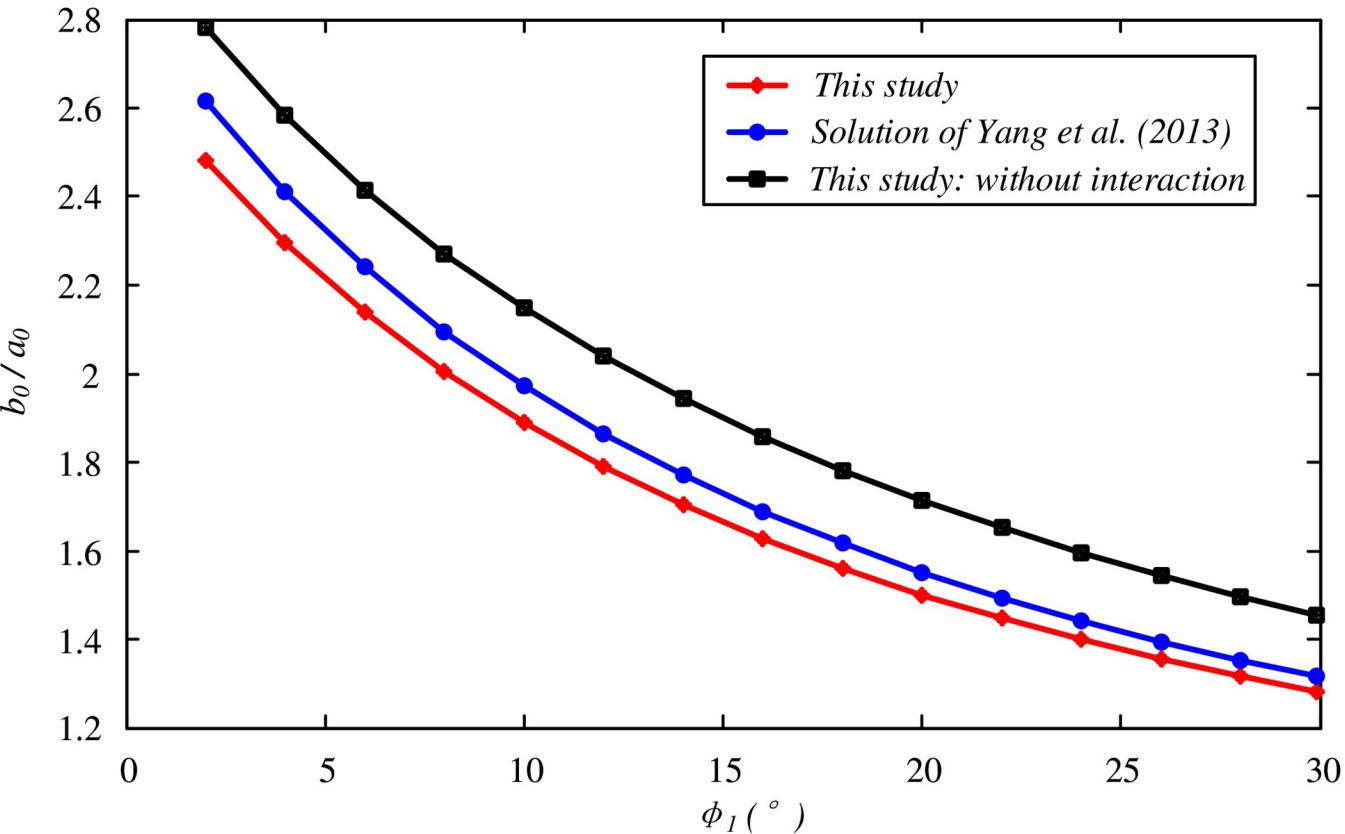

**Fig 9. Comparisons of optimal thickness of the FSW with Yang's solution and test without interaction effect.**

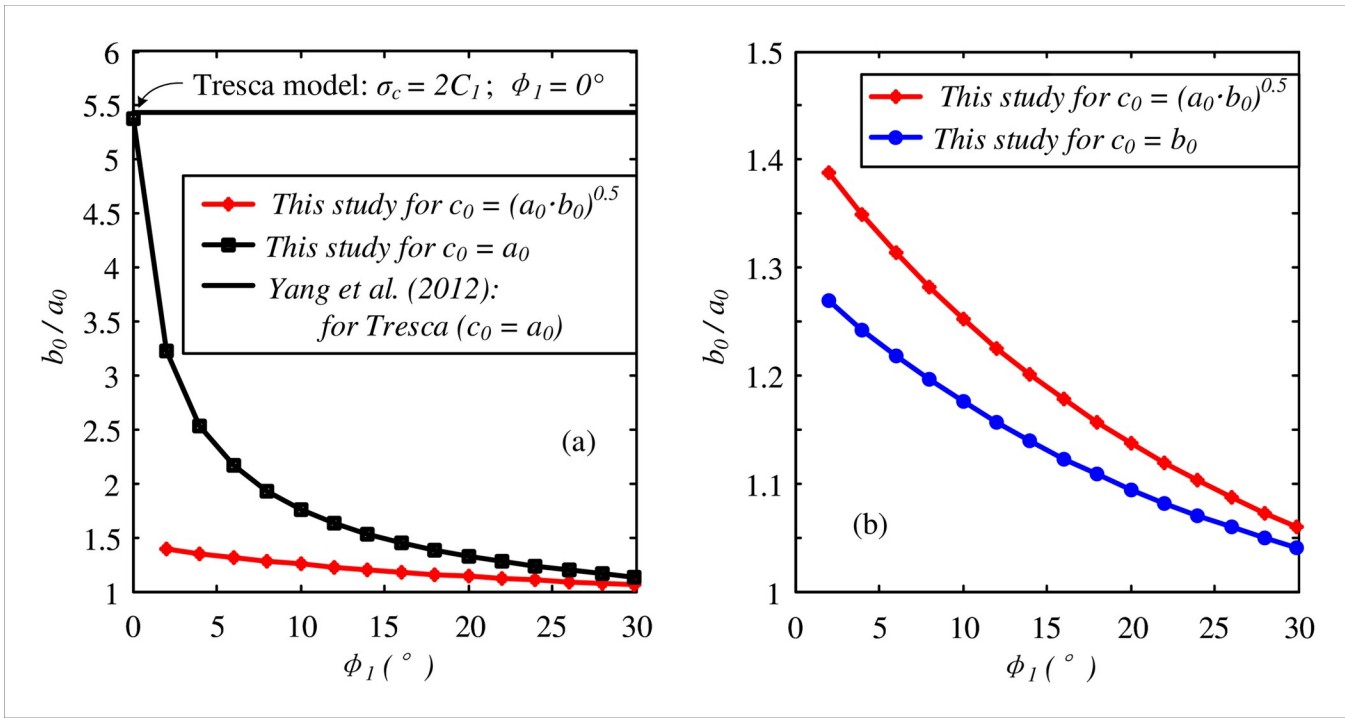

**Fig 10.** Comparisons with different critical conditions: (a) $c_0 = a_0$; (b) $c_0 = b_0$.

desired to determine the optimal thickness of FSW in terms of the design, and further research is required for investigating comparisons of the experimental work and field data.

Note that the temperature was assumed to be uniformly distributed across the frozen barrier, and that the elastic-perfectly plastic model was adopted to predict the behavior of the frozen soil. The initial hydrostatic stress assumption neglected the effect of stress in the third direction of the plane strain model and the influence of the stress distribution during the FSW formation. Although these assumptions lead to the limitations of the proposed analytical solution, the exact elastic-plastic analysis of the optimal thickness extended the existing design theory for the excavation of tunnels and shaft sinking using ground freezing technology and provided a benchmark for validating the numerical simulation, considering the interaction between the FSW and the surrounding soil.

## Conclusions

An elastic-plastic solution of the cavity contraction was developed in this paper to provide the optimal thickness of the frozen soil wall for excavation, considering the interaction between the frozen soil wall and the surrounding soil. The Mohr-Coulomb yield criterion with a non-associated flow rule was used to model the frozen soil, and a large strain assumption was adopted to account for the severe distortion of the soil element in the plastic region. Another novelty of this solution was the removal of the additional assumption of zero volumetric strain in the plastic region. A numerical simulation based on the finite element method was conducted to validate the solution regarding the distributions of stress, strains and displacements. The developed solution was then confirmed through good comparisons with the numerical simulation results. A thorough parametric study was provided by investigating the variation in the optimal thickness of the frozen soil wall with the soil properties, including the friction angle, dilation angle, cohesion, and Young's modulus, as well as the effect of the initial stress condition. The optimal thickness of the frozen soil wall based on the proposed solution was approximately 12% ~ 15% smaller than that of the model without an interaction. Additionally, a 2.5% ~ 5.5% reduction was found compared to the previous elastic-plastic solution, which may contribute to both the design and cost-effectiveness in practical engineering, such as tunnelling and mine shaft construction. The effect of the critical condition was also presented leading to a more reasonable elastic-plastic design theory for frozen soil walls.

## Author Contributions

**Conceptualization:** Lianfei Kuang.

**Formal analysis:** Bin Chen.

**Funding acquisition:** Lianfei Kuang, Pin-Qiang Mo, Kuan-Jun Wang.

**Investigation:** Lianfei Kuang.

**Methodology:** Pin-Qiang Mo, Kuan-Jun Wang.

**Resources:** Bin Chen.

**Software:** Bin Chen.

**Validation:** Kuan-Jun Wang.

**Writing – original draft:** Lianfei Kuang.

**Writing – review & editing:** Pin-Qiang Mo.

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
