## [Decision Letter · Decision Letter 0]

9 Mar 2022

PONE-D-22-04940An Elastic-plastic Solution for Optimal Thickness of Frozen Soil Wall Considering Interaction with Surrounding RockPLOS ONE

Dear Dr. Mo,

Thank you for submitting your manuscript to PLOS ONE. After careful consideration, we feel that it has merit but does not fully meet PLOS ONE’s publication criteria as it currently stands. Therefore, we invite you to submit a revised version of the manuscript that addresses the points raised during the review process.

We look forward to receiving your revised manuscript.

Kind regards,

Ahmed Salih Ahmed

Academic Editor

PLOS ONE

Journal Requirements:

"The authors would like to acknowledge financial supports from the National Natural Science Foundation of China (Grant No. 51908546, Grant No. 52178374, Grant No. 52108356), China Postdoctoral Science Foundation (Grant No. 2020T130699), and the Fundamental Research Funds for the Central Universities (Grant No.2020ZDPYMS18)."

"KUANG: the Fundamental Research Funds for the Central Universities (Grant No. 2020ZDPYMS18)

MO: National Natural Science Foundation of China (Grant No. 51908546, Grant No. 52178374), China Postdoctoral Science Foundation (Grant No. 2020T130699)

WANG: the National Natural Science Foundation of China (Grant No. 52108356)

Reviewers' comments:

Reviewer's Responses to Questions

**Comments to the Author**

1. Is the manuscript technically sound, and do the data support the conclusions?

Reviewer #1: Yes

Reviewer #2: Yes

2. Has the statistical analysis been performed appropriately and rigorously? 

Reviewer #1: N/A

Reviewer #2: I Don't Know

3. Have the authors made all data underlying the findings in their manuscript fully available?

Reviewer #1: Yes

Reviewer #2: No

4. Is the manuscript presented in an intelligible fashion and written in standard English?

Reviewer #1: Yes

Reviewer #2: No

5. Review Comments to the Author

Reviewer #1: 1. Establish a specific research goal.

2. Clearly define the type of soil used in the analysis using one of the commonly used classification systems.

3. What is the numerical analysis technique or program?

4. The cohesion of the soil used in the analysis has a very high value.

5. The modulus of elasticity of the soil used in the analysis is exceptionally high.

6. E1 = 300 mPa for FSW in line 222, and E2 = 300 mPa for surrounding in line 223. How?

7. Line 271, "C1 variation between 0.35 and 0.35 MPa." How?

8. Extend the discussion by mentioning the numerical results in conclusion.

9. paraphrasing a few sentences as shown in the attached file

Reviewer #2: Dear authors,

The following points need revisions:

1- The abstract should state the major points of your research briefly and explain why your work is important, your purpose, how you went about your project, what you learned, and what you concluded. An abstract is often presented separately from the article, so it must be able to stand alone.

2- The introduction serves the purpose of leading the reader from a general subject area to a particular field of research. It establishes the context of the research being conducted by summarizing current understanding and background information about the topic, stating the purpose of the work in the form of the hypothesis, question, or research problem, briefly explaining your rationale, methodological approach, highlighting the potential outcomes your study can reveal, and describing the remaining structure of the paper.

3- Remove any the figures from the introduction.

4- A flowchart should be added to the article to show the research methodology.

5- The necessity and innovation of the article should be presented to the introduction.

6- The paper language and punctuation need revision by an expert.

7- Sections 2 Problem definition and 3 Analytical solution of cavity contraction, should be subsection under the heading "Methodology".

8- Enhancing the study with realistic images of experimental work.

9- Consider enhancing the used references as they are few and old.

10- Add the DOI to all references.

6. PLOS authors have the option to publish the peer review history of their article (what does this mean?). If published, this will include your full peer review and any attached files.

Reviewer #1: No

Reviewer #2: **Yes: **Laheab A. Almaliki

---

## [Author Response · Author response to Decision Letter 0]

25 Mar 2022

Please check the file 'Response to Reviewers' for more information. The same text is attached below:

First of all, we would like to express our sincere thanks to the reviewers for their constructive comments and thoughtful suggestions. The following is a point-to-point response to the reviewers’ comments. Note that the corresponding line numbers in the revised manuscript with track changes have been updated and marked by colored font according to the reviewers’ advice. The reds are corrections by the authors, and the greens are from the scientific editing by AJE.

Reviewer #1: 

[Comment 1-1]:

Establish a specific research goal.

[Response 1-1]:

The abstract and the section of Introduction have been modified according to the reviewers’ suggestions, to improve the expressions, with noting the research purpose of this study: “to provide the optimal thickness of the frozen soil wall for excavation using technology of artificial ground freezing” and “a rigorous elastic-plastic solution of cavity contraction is proposed using a non-associated Mohr-Coulomb failure criterion, …, with considerations of the interaction between frozen soil wall and the surrounding soil/rock”.

[Comment 1-2]:

Clearly define the type of soil used in the analysis using one of the commonly used classification systems.

[Response 1-2]:

It is noted that the proposed solution is applicable to all soil types, as long as the behaviour of soil or frozen soil can be captured by Mohr-Coulomb criterion. This comment has been added to the end of Section3, with “It is noted that the proposed solution is applicable to all soil types, and the mechanical responses of the unfrozen and frozen materials are characterized by the Mohr-Coulomb failure criterion.”.

As to the case study in Section 4, the sinking shaft in Tertiary clay is adopted for analysis. According to the Unified Soil Classification System, the original soil belongs to CL for clay soil with low plasticity. Related information has also been provided in the text, with “According to the Unified Soil Classification System, the soil can be termed as clay soil with low plasticity (CL).”.

[Comment 1-3]:

What is the numerical analysis technique or program?

[Response 1-3]:

The proposed solution was formulated based on Matlab for calculation, while the numerical simulation using Abaqus was also conducted to validated the analytical solution , as shown in Fig. 3. Some relevant information have been provided as: “A Matlab-based code is then programmed to calculate the distributions of stresses and strains after cavity contraction, and to provide the optimal thickness of the frozen soil wall for excavation.” and “Numerical simulation using Abaqus-based finite element model with the calculated optimal thickness of FSW was conducted for validation of the proposed analytical solution”.

[Comment 1-4]:

The cohesion of the soil used in the analysis has a very high value.

[Response 1-4]:

The cohesion value (3.5 MPa) in this paper is for the frozen soil at -20 ℃, based on data of experimental tests from the literature (Yang et al, 2013), rather than typical unfrozen soil. That’s why the value is relatively high. 

Based on data from the following references, this value can be considered as normal.

1. Paper ‘Experimental Study of Frozen Soil Mechanical Properties for Seismic Design of Pile Foundations’ from ‘Cold Regions Engineering 2012’: ultimate strength of frozen soil is about 3.2 MPa. 

2. Paper ‘Mechanical properties of seasonally frozen and permafrost soils at high strain rate’ from ‘Cold Regions Science and Technology’: ultimate strength of frozen soil (T<10 ℃) is between 2.3~7.1 MPa.

Relevant comments have been provided in the text to explain the high value of cohesion.

[Comment 1-5]:

The modulus of elasticity of the soil used in the analysis is exceptionally high.

[Response 1-5]:

Again, the properties of frozen and unfrozen soils are based on data of experimental tests from the literature (Yang et al, 2013). It is noted that the elastic modulus of unfrozen soil was noted as ‘300 MPa’ by mistake. It should be corrected to ‘100 MPa’, which was used in the following calculation. Compared to the typical value of Young’s modulus (30~60 MPa) for clays with low plasticity (CL), it is high. However, the soil in the case study is embedded in 500 m depth, and the large confining stress results in high Young’s modulus for compacted soils. It is therefore reasonable to get 100 MPa of elastic modulus for unfrozen soil. Some references also show larger ranges:

The elastic modulus of frozen soil is ‘300 MPa’, which is reasonable for soil at -20 ℃. By checking some relevant literature, we can easily find similar value of elastic modulus. For example, Paper ‘Experimental Study of Frozen Soil Mechanical Properties for Seismic Design of Pile Foundations’ from ‘Cold Regions Engineering 2012’ reported the Young’s modulus of frozen soil varies between 270 MPa and 320 MPa. 

[https://ascelibrary.org/doi/epdf/10.1061/9780784412473.047]

Relevant comments have been provided in the text to explain the high value of modulus.

[Comment 1-6]:

E1 = 300 mPa for FSW in line 222, and E2 = 300 mPa for surrounding in line 223. How?

[Response 1-6]:

We feel sorry for this typing mistake. The elastic modulus of frozen soil is 300 MPa, while the magnitude for unfrozen soil is 100 MPa. These values have been explained in the last question, and E2 value has been corrected in the revised paper.

[Comment 1-7]:

Line 271, "C1 variation between 0.35 and 0.35 MPa." How?

[Response 1-7]:

Thanks for pointing out this mistake. It was also a typo-mistake, and the correct range should be between 0.35 and 35 MPa, as revised in the paper. 

[Comment 1-8]:

Extend the discussion by mentioning the numerical results in conclusion.

[Response 1-8]:

Thanks for your suggestion. The conclusion has been revised with additional discussion on numerical simulation: “Numerical simulation based on Finite Element Method was conducted to validate the solution regarding to the distributions of stress, strains and displacements. The developed solution was then confirmed by the good comparisons with results of numerical simulation.”. 

[Comment 1-9]:

paraphrasing a few sentences as shown in the attached file

[Response 1-9]:

Although I have not received any attached file from the Journal, we have revised the technical writing of this paper thoroughly. Hope the revised paper with the current version has met all requirements for publication.

Reviewer #2: 

[Comment 2-1]:

The abstract should state the major points of your research briefly and explain why your work is important, your purpose, how you went about your project, what you learned, and what you concluded. An abstract is often presented separately from the article, so it must be able to stand alone.

[Response 2-1]:

The abstract of this paper has been modified substantially, according to the constructive suggestion. 

[Comment 2-2]:

The introduction serves the purpose of leading the reader from a general subject area to a particular field of research. It establishes the context of the research being conducted by summarizing current understanding and background information about the topic, stating the purpose of the work in the form of the hypothesis, question, or research problem, briefly explaining your rationale, methodological approach, highlighting the potential outcomes your study can reveal, and describing the remaining structure of the paper.

[Response 2-2]:

Thanks for your constructive suggestion. The section of Introduction has been modified accordingly, especially to strengthen the statements about the background, limitations and the purposes of this study. Also, the novelty of this study has been introduced and the general structure of this paper has been briefly described.

[Comment 2-3]:

Remove any the figures from the introduction.

[Response 2-3]:

The original Fig. 1 in the section of Introduction has been removed.

[Comment 2-4]:

A flowchart should be added to the article to show the research methodology.

[Response 2-4]:

A flowchart of the analytical solution has been added for calculation of optimal thickness of FSW.

[Comment 2-5]:

The necessity and innovation of the article should be presented to the introduction.

[Response 2-5]:

Thanks for pointing out this flaw of the manuscript. In the revised Introduction, the drawbacks of the existing solutions have been further explained to highlight the necessity of this study. It is now noted as: “However, the current solutions neglect the effects of friction induced dilation, which is typically significant to the geomaterials. Large deformation caused by excavation is usually assumed as infinitely small, that could result in large errors in the estimation of optimal thickness of the FSW. The outer confining stress of the FSW is a changing parameter during construction, and plays a vital role in the FSW-soil interaction system. It is therefore necessary to improve the analytical method with considerations of both cohesive-frictional behaviour of frozen soil and interactions between the FSW and the surrounding soil.”.

The innovation of this study has also been strengthened in the last paragraph of Introduction. Some additional comments are: “This paper attempts to provide a rigorous elastic-plastic solution of cavity contraction for the design of optimal thickness of the frozen soil wall with consideration of interaction with the surrounding soil.”; “For soil in the plastic region, large strain analysis is incorporated within the solution to account for the excavation induced large deformation, while the assumption of no volumetric strain is eliminated.”; “The innovation of this study is also instantiated by the proposed optimal thickness of the frozen soil wall”.

[Comment 2-6]:

The paper language and punctuation need revision by an expert.

[Response 2-6]:

The language of this paper has been revised by a native speaking expert. Additionally, the suggested professional scientific editing service helps to improve the writing of this paper, with green corrections. Hope it is better now to be accepted for publication.

[Comment 2-7]:

Sections 2 Problem definition and 3 Analytical solution of cavity contraction, should be subsection under the heading "Methodology".

[Response 2-7]:

The structure of this paper has been modified according to the suggestion. Section of “Methodology” has been added to merge ‘Problem definition’ and ‘Analytical solution’.

[Comment 2-8]:

Enhancing the study with realistic images of experimental work.

[Response 2-8]:

As this study focuses on the analytical study, and no experimental work was conducted specifically for this study. All parameters were selected based on previous studies from literature. Comments have been provided to explain their reasonability. 

However, a case study was used to set the parameters in Section 3, and the background of this case study has been further introduced, with typical experimental results of a sample from this site. Hope this additional figure could assist the understanding on the basic mechanical response of frozen soil at high stress condition. 

[Comment 2-9]:

Consider enhancing the used references as they are few and old.

[Response 2-9]:

The references of this paper have been enhanced as suggested. Some recent studies related to AGF and frozen soils have been included.

[Comment 2-10]:

Add the DOI to all references.

[Response 2-10]:

Thanks for your suggestion. DOI for all references have been added, if available. DOI for some papers are not available, and ISBN numbers are given instead for textbooks.

---

## [Decision Letter · Decision Letter 1]

1 Apr 2022

An elastic-plastic solution for the optimal thickness of a frozen soil wall considering an interaction with the surrounding rock

PONE-D-22-04940R1

Dear Dr. Mo,

We’re pleased to inform you that your manuscript has been judged scientifically suitable for publication and will be formally accepted for publication once it meets all outstanding technical requirements.

Kind regards,

Ahmed Mohammed

Academic Editor

PLOS ONE

Additional Editor Comments (optional):

Reviewers' comments:

Reviewer's Responses to Questions

**Comments to the Author**

1. If the authors have adequately addressed your comments raised in a previous round of review and you feel that this manuscript is now acceptable for publication, you may indicate that here to bypass the “Comments to the Author” section, enter your conflict of interest statement in the “Confidential to Editor” section, and submit your "Accept" recommendation.

Reviewer #1: All comments have been addressed

Reviewer #2: All comments have been addressed

2. Is the manuscript technically sound, and do the data support the conclusions?

Reviewer #1: Yes

Reviewer #2: Yes

3. Has the statistical analysis been performed appropriately and rigorously? 

Reviewer #1: N/A

Reviewer #2: Yes

4. Have the authors made all data underlying the findings in their manuscript fully available?

Reviewer #1: Yes

Reviewer #2: No

5. Is the manuscript presented in an intelligible fashion and written in standard English?

Reviewer #1: Yes

Reviewer #2: Yes

6. Review Comments to the Author

Reviewer #1: I would introduce my thank the author for the wonderful efforts in preparing the research, taking all the previous notes, and working to correct the errors

Reviewer #2: (No Response)

7. PLOS authors have the option to publish the peer review history of their article (what does this mean?). If published, this will include your full peer review and any attached files.

Reviewer #1: **Yes: **Maki Jafar M. Al-Waily

Reviewer #2: No

---

## [Editor Report · Acceptance letter]

14 Apr 2022

PONE-D-22-04940R1 

An elastic-plastic solution for the optimal thickness of a frozen soil wall considering an interaction with the surrounding rock 

Dear Dr. Mo:

I'm pleased to inform you that your manuscript has been deemed suitable for publication in PLOS ONE. Congratulations! Your manuscript is now with our production department. 

Kind regards, 

on behalf of

Dr. Ahmed Mohammed 

Academic Editor

PLOS ONE